# Yeast Hydrolysate and Postmenopausal Osteoporosis

**DOI:** 10.3390/jpm13020322

**Published:** 2023-02-13

**Authors:** Yang Hee Hong, Eun Young Jung

**Affiliations:** 1Department of Beauty and Art, Suwon Women’s University, Suwon 441-748, Republic of Korea; 2Department of Home Economic Education, Jeonju University, Jeonju 560-759, Republic of Korea

**Keywords:** bone, ovariectomy, postmenopausal osteoporosis, yeast hydrolysate, rats, osteocalcin, estrogens

## Abstract

We used an ovariectomy (OVX) rat model to test whether yeast hydrolysate (YH) has therapeutic effects on postmenopausal osteoporosis-induced bone loss. The rats were separated into five treatment groups: the sham group (sham operation); the control group (no treatment after OVX); the estrogen group (estrogen treatment after OVX); YH 0.5% group (drinking water supplementation with 0.5% YH after OVX); and the YH 1% group (drinking water supplementation with 1% YH after OVX). In addition, the YH treatment restored serum testosterone concentration in the OVX rats up to the normal level. Further, YH treatment affected bone markers; a significant increase in serum calcium concentration was observed after adding YH to the diet. The levels of serum alkaline phosphatase, osteocalcin, and cross-linked telopeptides of type I collagen were reduced by YH supplementation, unlike those in the no-treatment control. Although not statistically significant, YH treatment in OVX rats improved trabecular bone microarchitecture parameters. These results show that YH may ameliorate the bone loss caused by postmenopausal osteoporosis because of the normalization of serum testosterone concentration.

## 1. Introduction

Osteoporosis weakens bones to the point that they can break easily [1]. It is a worldwide health problem with high prevalence not only in Western countries but also in Asia [2]. Osteoporosis associated with an ovarian hormone deficiency after menopause (postmenopausal osteoporosis, PO) is the most common cause of age-related bone loss [3]. A therapeutic regimen used to treat PO and found to have some modulatory effect on bone loss prevents ovariectomy (OVX)-induced bone loss [4]. The OVX rat model remains the most popular animal model because it has been validated to reproduce the most important clinical features of estrogen deficiency in humans: induced bone loss, (PO) [5]. The Food and Drug Administration recommends the OVX rat model for the preclinical evaluation of new therapeutic candidates for PO [6].

Additionally, estrogen replacement therapy has been recognized as a possible remedy for PO-related problems [7,8]. Although estrogen is known to decrease the risk of PO and improve bone turnover markers [9,10], it is well known to be associated with a higher risk of hormone-related cancer [11,12] and other adverse effects [13]. Further, estrogen replacement therapy is thus recommended for women at a high risk of PO with no contraindications [14]. Therefore, it would be most helpful to discover a naturally occurring substance that minimizes bone loss in PO, thereby decreasing the need for hormone therapy.

In previous studies, we found that yeast hydrolysate (YH) effectively promotes bone metabolism in vitro and in vivo [15,16]. We reported that YH increases osteoblast proliferation and directly upregulates the expression of alkaline phosphatase (ALP) and bone matrix proteins; these changes trigger osteoblastic differentiation [15]. We also confirmed that YH promotes longitudinal bone growth in adolescent rats [16]. YH can relieve discomfort or bone damage in postmenopausal women and can be a useful natural supplement in PO treatment. To test this hypothesis, we used the OVX rat model to determine whether YH has therapeutic effects on bone loss caused by PO.

## 2. Materials and Methods

### 2.1. Preparation of YH

Saccharomyces cerevisiae IFO 2346 was cultured for 3 days at 30 °C in a medium containing 2% molasses, 0.6% (NH_4_)_2_SO_4_, 0.1% MgSO_4_·7H_2_O, 0.2% KH_2_PO_4_, 0.03% K_2_HPO_4_, and 0.1% NaCl at 30 °C for 3 days. After incubation, the culture was centrifuged at 10,000× *g* for 20 min to harvest the cells. Cells were resuspended in 20 mM phosphate buffer (pH 7.0) and hydrolyzed with 1000 units of bromelain at 30 °C for 4 h. The hydrolysate was subsequently centrifuged at 10,000× *g* for 20 min. The hydrolysate was passed through a membrane (Sartocon cassette, Sartorius, Göttingen, Germany) to obtain the 10–30 kDa peptide fraction. This fraction was used as YH in this study.

### 2.2. Animals and Diets

Thirty-five 10-week-old female Sprague–Dawley rats were purchased from Nara Biotech Inc. (Seoul, Republic of Korea). The rats were kept in individual wire-floor cages with a 12-h light-dark cycle in a room without windows. The rats were fed a commercial rodent diet (Samyang Co., Seoul, Republic of Korea) with the following composition (grams per kilogram of diet): moisture, 80; protein, 230; fat, 35; fiber, 50; and carbohydrate, 600. The feed and water were provided *ad libitum*. The rats were randomly divided into five treatment groups after a week of acclimation. (n = 7): sham group, sham operation; control group, no treatment after OVX; estrogen group, estrogen treatment after OVX; YH 0.5% group, drinking water supplementation with 0.5% YH after OVX; and YH 1% group, drinking water supplementation with 1% YH after OVX. The details of the procedures (sham and OVX operations) have been previously described [17]. Sesame oil was combined with estrogen after it had been dissolved in a small amount of absolute ethanol in a ratio of 97% corn oil to 3% ethanol. Using subcutaneous injections (10 μg·kg^−1^ BW), estrogen was given. Based on daily water consumption, YH at 0.5% and 1% in drinking water corresponds to doses of 1 and 2 g·kg^−1^ BW, respectively. Weekly checks were made on both food intake and body weight.

### 2.3. Biochemical Analyses

Cardiovascular puncture was used to obtain blood samples, and centrifugation at 3000× *g* for 15 min was used to separate the serum. Serum levels of estradiol, growth hormone, growth hormone-releasing factor, osteocalcin, and cross-linked telopeptide of type I collagen (CTx) were measured using commercially available enzyme-linked immunosorbent assays (USCN Life Science Inc., Beijing, China), and those of testosterone were measured using the ELISA kit from R&D System Inc. (Minneapolis, MN, USA). Serum calcium and ALP levels were determined using an automatic blood analyzer (Fuji Photo Film Co., Osaka, Japan). After blood collection, organs such as the liver, kidney, spleen, and uterus were removed, weighed immediately on an electronic scale, and checked for gross abnormalities.

### 2.4. Microcomputed Tomography (μ-CT) Bone Analyses

μ-CT scans (SkyScan 1072, SkyScan, Kontich, Belgium) were performed on Ward’s triangle of the rat femur at a tube potential (peak) of 80 KVp, an intensity of 100 μA, and an integration time of 3400 ms. We used the manufacturer’s software package (SkyScan CT analyzer, SkyScan, Kontich, Belgium) for analyzing the images and calculating the bone indices. The bone indices assessed were the bone volume fraction (bone volume [BV]/trabecular volume [TV], BV/TV), trabecular number (Tb.N), trabecular thickness (Tb.Th), and trabecular separation (Tb.Sp).

### 2.5. Statistical Analyses

The Statistical Package for Social Sciences (SPSS), version 12.0 (IBM Corporation, Armonk, NY, USA), was used to analyze the data. For each response variable, a one-way analysis of variance (ANOVA) with Tukey’s multiple-range test was run at the *p* < 0.05 level of significance.

## 3. Results and Discussion

By measuring uterine weight at euthanasia, it was confirmed that the animals had a successful OVX. [18]. Compared to the sham-operated animals, the OVX animals had significantly smaller uteruses (0.77 g for OVX rats versus 0.40 g for sham-operated rats, *p* < 0.05), without significant differences in uterine weight among OVX animals (data not shown). The YH treatment had no effect on the rats’ relative internal organ weights (liver, kidney, and spleen). Furthermore, there were no discernible abnormalities in the internal organs of the experimental rats upon gross examination (data not shown). The internal organs were not harmed by the YH treatment.

There was a significant difference in body weight between OVX rats and sham-operated rats (Figure 1). OVX significantly increased body weight (*p* < 0.05). Estrogen treatment significantly attenuated the body weight gain during 8 weeks (*p* < 0.05), and this effect of estrogen treatment was diminished at week 8. No significant differences in body weight gain were observed between YH-treated and untreated OVX rats. There were no significant differences in food intake between the OVX and sham-operated rats. Estrogen and YH treatment did not affect food intake in OVX rats (data not shown).

The changes in serum hormone levels after the 8-week YH treatment are shown in Figure 2. A significant increase in serum testosterone concentration was observed after adding YH to the diet (*p* < 0.05; Figure 2A). The decrease in serum testosterone concentration was significantly attenuated by the YH supplementation (*p* < 0.05); this treatment restored the serum testosterone level in OVX rats to normal. Nonetheless, a dose-dependent relationship was not observed; there was no significant difference between groups YH 0.5% and YH 1%. OVX did not affect serum concentrations of estradiol, growth hormone, or growth hormone-releasing factor. There were no significant differences between the treatment groups (estrogen and YH treatment) and the no-treatment group of OVX rats (Figure 2B–D).

OVX resulted in changes in serum levels of bone metabolism biomarkers. Figure 3 shows the serum biomarkers of the bone metabolism of OVX rats treated with YH for 8 weeks. The serum calcium level was significantly reduced by OVX (*p* < 0.05) and was significantly elevated up to the normal level by YH and the estrogen treatment (*p* < 0.05; Figure 3A). The serum ALP level was significantly increased by OVX (*p* < 0.05) and was significantly decreased by the 0.5% YH treatment (*p* < 0.05; Figure 3B). The serum osteocalcin level was also increased by OVX and was significantly reduced by the 0.5% YH treatment (*p* < 0.05; Figure 3C). The serum CTx concentration was significantly increased by OVX. Meanwhile, estrogen and YH treatment significantly decreased the serum CTx level to normal (*p* < 0.05; Figure 3D).

The trabecular bone microarchitecture according to μ-CT is shown in Table 1. No significant differences in BV/TV and Tb.Th were observed between OVX and sham-operated rats. Estrogen and YH treatment of OVX rats improved BV/TV, Tb.N, and Tb.Sp, without any statistical significance.

According to Wronski et al. [19], bone loss can be seen in rodents as early as two weeks after OVX, and a linear loss can be seen for up to one hundred days following the operation. Additionally, bone formation starts to significantly decline 30 days after surgery, so we began our dietary intervention at that time and kept it up for 8 weeks. This strategy was intended to give the animals enough time to recover from the procedure and to make it possible to monitor any potential advantages of YH for preventing bone loss during menopause. A recent study by Dang et al. [20] suggests that estrogen plays a significant role in stimulating the differentiation of progenitor cells via the osteoblast lineage, even though the precise mechanisms by which OVX increases body weight are unknown. In a rat model, body fat increases in the absence of estrogen, increasing body weight, as seen in the OVX control group.

Testosterone in postmenopausal women can be a useful marker of certain postmenopausal symptoms [21]. Postmenopausal women, particularly those after OVX, have consistently low testosterone levels [22]. The decline in testosterone levels in these women is associated with pathological changes, including atherosclerosis, tiredness, impairment of libido, and PO [23]. In this study, serum testosterone concentration was low in OVX rats, and YH treatment significantly elevated the level. Therefore, it is a useful finding that YH seems to normalize serum testosterone in PO-affected OVX rats.

Remodeling is necessary for healthy bones. It is a tightly interconnected process (formation is linked to resorption). Osteoclasts start the process by resorbing old bones, and then osteoblasts create new bones [4]. Due to resorption outpaces formation, bone loss begins after middle age or possibly earlier. A lack of estrogen exacerbates this imbalance. Resorption markers and formation markers are two types of biochemical markers that can be monitored and reflect remodeling [24].

In order to clarify the effects of YH on bone loss related to PO, we analyzed bone formation markers such as serum calcium, ALP, and osteocalcin and a bone resorption marker (serum CTx) in OVX rats treated with YH. The results showed that YH treatment affected bone markers; a significant increase in serum calcium concentration was observed after adding YH to the diet, whereas serum ALP, osteocalcin, and CTx levels were reduced by YH treatment, unlike in the no-treatment group. According to our previous study [15], YH treatment upregulates ALP and bone matrix proteins such as bone sialoprotein (BSP) and collagen type II (COL II) in osteoblasts.

As a result of the pathological observation of the tissue, histological cystic lesions due to ovariectomy, conjugated estrogens, and yeast hydrolysates were not observed. The H&E-stained brain tissue was observed under an optical microscope at a magnification of 400, and pathologically acceptable necrosis and desquamation were not observed in the tissues of all groups (Figure 4).

In order to describe the trabecular compartment’s bone structure, the following morphometric variables were measured: BV/TV, a predictor of bone strength, and Tb.Th, the average thickness of the cancellous bone structure [25]. Furthermore, Tb.N and Tb.Sp are measures of the number of trabecular plates per unit of length and the average diameter of the marrow cavities, respectively [26,27]. Due to the fact that trabecular bones are more readily lost due to OVX [28], it is reasonable to assume that the trabeculae are more responsive to treatment. According to other researchers’ findings [29], a CT scan of the trabecular bones reveals that OVX significantly lowers Tb.N while raising Tb.Sp. Our findings suggest that YH treatment may counteract the negative effects of OVX. Restoring Tb.N. is a crucial step toward enhancing bone strength. However, due to the small sample size (possibly), the data did not reach statistical significance. The rise in Tb.N could be the cause of the decline in Tb.Sp. After OVX, Tb.Th has been shown to decrease in several studies [30,31]. In contrast to those results, Tb.Th was not decreased in the OVX in this study, which is consistent with the study by Devareddy et al. [32]. We can assume that the differences in age and number of days after OVX between the animals are the cause of this discrepancy.

## 4. Conclusions

YH affects bone parameters that are used as PO markers; these results suggest that by normalizing the serum testosterone level, YH may affect both bone formation markers (serum calcium, ALP, and osteocalcin) and bone resorption markers (e.g., serum CTx) in relation to PO. Our results indicate that YH is a promising nutraceutical agent for PO treatment; however, further research is needed to determine the effects of YH on trabecular bone microarchitecture and to identify the bioactive ingredient(s) of YH.

## Figures and Tables

**Figure 1 jpm-13-00322-f001:**
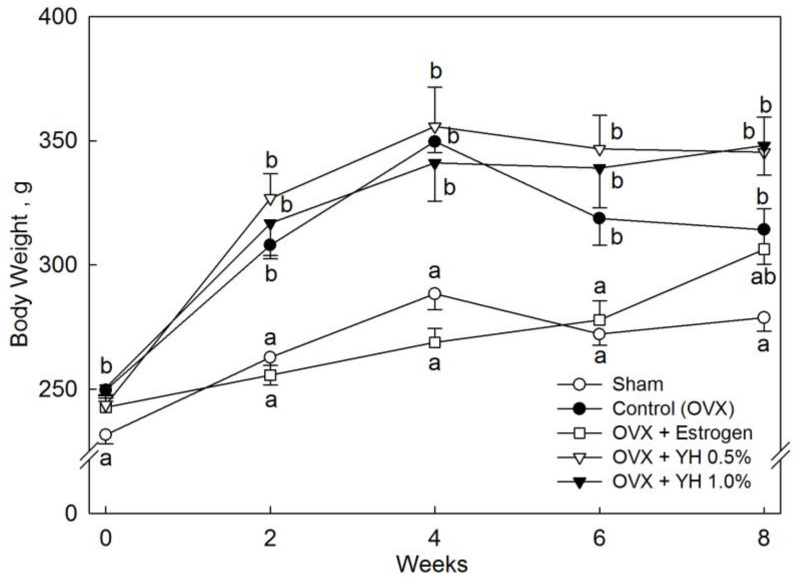
Body weight of ovariectomized (OVX) rats treated with yeast hydrolysate (YH) for 8 weeks. The data are presented as mean ± SEM for seven rats. According to Tukey’s multiple range tests, means with different superscript letters are significantly different (*p* < 0.05). Sham: sham operation; Control: no treatment after OVX; Estrogen: estrogen treatment after OVX; YH, 0.5%: drinking water supplementation with 0.5% YH after OVX; YH, 1%: drinking water supplementation with 1% YH after OVX. YH 0.5% and 1% in drinking water are equivalent to a dose of 1 and 2 g·kg^−1^ BW, respectively, based on daily water consumption.

**Figure 2 jpm-13-00322-f002:**
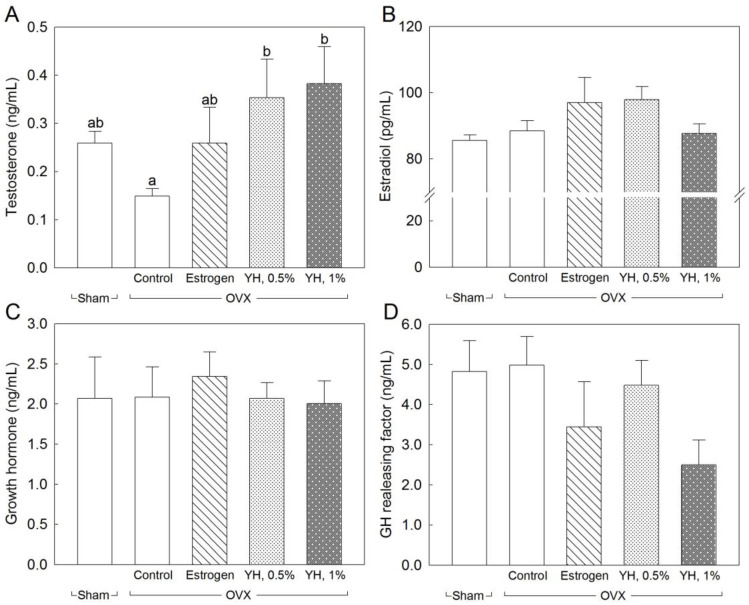
Serum hormone levels of ovariectomized (OVX) rats treated with yeast hydrolysate (YH) for 8 weeks. The data are presented as mean ± SEM for seven rats. According to Tukey’s multiple range tests, means with different superscript letters are significantly different (*p* < 0.05). (**A**): testosterone; (**B**): estradiol; (**C**): growth hormone; (**D**): growth hormone releasing factor. Sham: sham operation; Control: no treatment after OVX; Estrogen: estrogen treatment after OVX; YH, 0.5%: drinking water supplementation with 0.5% YH after OVX; YH, 1%: drinking water supplementation with 1% YH after OVX. YH 0.5% and 1% in drinking water are equivalent to a dose of 1 and 2 g·kg^−1^ BW, respectively, based on daily water consumption.

**Figure 3 jpm-13-00322-f003:**
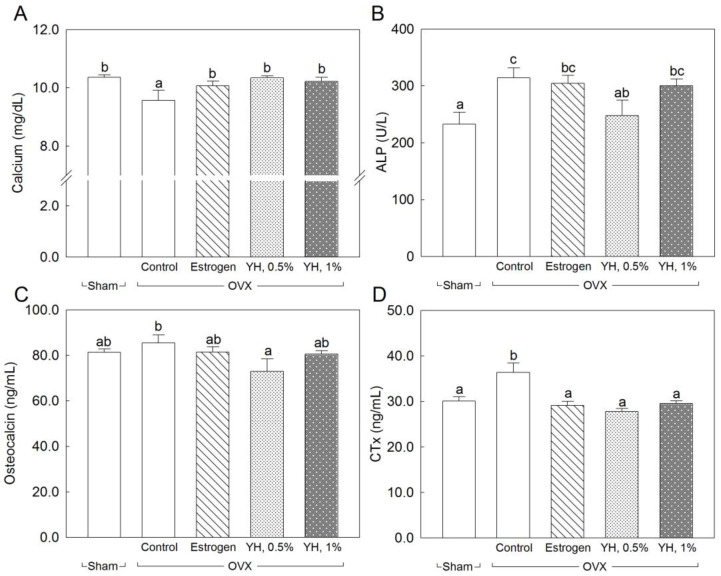
Serum biomarkers of bone metabolism of ovariectomized (OVX) rats treated with yeast hydrolysate (YH) for 8 weeks. The data are presented as mean ± SEM for seven rats. According to Tukey’s multiple range tests, means with different superscript letters are significantly different (*p* < 0.05). (**A**): calcium; (**B**): alkaline phosphatase (ALP); (**C**): osteocalcin; (**D**): cross-linked telopeptide of type I collagen (CTx). Sham: sham operation; Control: no treatment after OVX; Estrogen: estrogen treatment after OVX; YH, 0.5%: drinking water supplementation with 0.5% YH after OVX; YH, 1%: drinking water supplementation with 1% YH after OVX. YH 0.5% and 1% in drinking water are equivalent to a dose of 1 and 2 g·kg^−1^ BW, respectively, based on daily water consumption.

**Figure 4 jpm-13-00322-f004:**
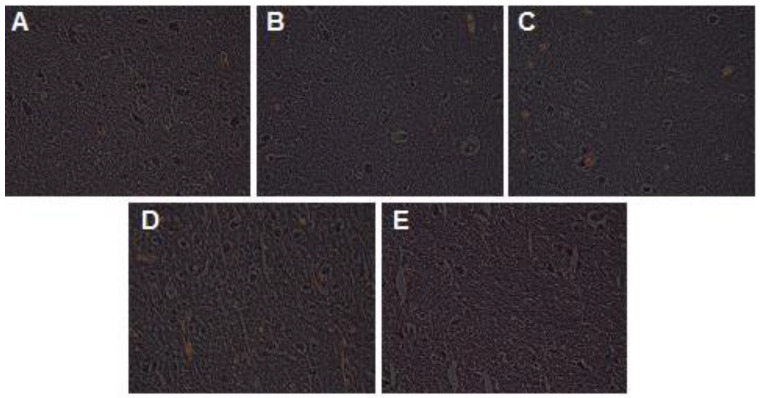
Representative microscopic findings in brain of ovariectomized (OVX) rats treated with yeast hydrolysate (YH) for 8 weeks (hematoxylin-eosin stain, ×400). (**A**) Sham: sham operation; (**B**) Control: no treatment after OVX; (**C**) Estrogen: estrogen treatment after OVX; (**D**) YH, 0.5%: drinking water supplementation with 0.5% YH after OVX; (**E**) YH, 1%: drinking water supplementation with 1% YH after OVX. YH 0.5% and 1% in drinking water are equivalent to a dose of 1 and 2 g·kg^−1^ BW, respectively, based on daily water consumption.

**Table 1 jpm-13-00322-t001:** Trabecular bone microarchitecture of ovariectomized (OVX) rats treated with yeast hydrolysate (YH) for 8 weeks.

Parameters	Sham	Control	OVX	OVX	OVX	Pooled SEM
			Estrogen	YH, 0.5%	YH, 1%	
BV/TV, %	44.38 ^b^	38.79 ^ab^	44.63 ^b^	37.89 ^ab^	36.16 ^a^	6.21
Tb.N, 1/mm	10.75 ^c^	7.14 ^a^	7.40 ^ab^	8.62 ^bc^	7.96 ^ab^	1.91
Tb.Th, mm	57.38	54.05	63.92	47.10	46.27	8.32
Tb.Sp, mm	54.19 ^a^	82.32 ^b^	77.37 ^ab^	76.98 ^ab^	81.73 ^b^	10.12

The data are presented as mean ± SEM for seven rats. According to Tukey’s multiple range tests, means with different superscript letters are significantly different (*p* < 0.05). Sham: sham operation; Control: no treatment after OVX; Estrogen: estrogen treatment after OVX; YH, 0.5%: drinking water supplementation with 0.5% YH after OVX; YH, 1%: drinking water supplementation with 1% YH after OVX. YH 0.5% and 1% in drinking water are equivalent to a dose of 1 and 2 g·kg^−1^ BW, respectively, based on daily water consumption.

## Data Availability

The data that support the findings of this study are available from the Hong, Y.H. or Jung, E.Y upon reasonable request.

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
