# Peer review of "Yeast Hydrolysate and Postmenopausal Osteoporosis"

_jpm, 2023, doi:10.3390/jpm13020322_

Round 1

Reviewer 1 Report

Hong et al. used ovariectomy (OVX) rat model to study the effects of yeast hydrolysate (YH) on postmenopausal osteoporosis-induced bone loss. The authors revealed that lowered serum testosterone level in the OVX rats was elevated significantly by YH treatment. As a result, bone formation markers were increased, bone resorption markers were reduced, and parameters of trabecular bone microarchitecture was improved. The manuscript is well written and organized. I have no further comments.

Author Response

Thank you for your review.

Reviewer 2 Report

The authors used a rat model of ovariectomy to test whether yeast hydrolyzate has therapeutic effects on bone loss induced by postmenopausal osteoporosis. Although not statistically significant, the results show that treatment with yeast hydrolyzate restored the serum testosterone concentration in the ovariectomy rats to the normal level. This study brings good pre-clinical results and will certainly be the basis for new clinical studies on the subject of osteoporosis.

Author Response

Thank you for your review.

Reviewer 3 Report

It is necessary to consider the following aspects:

1.-about title of the work... Study of the effect of yeast hydrolyzate and osteoporosis postmenopausal in an model experimental.

2.-Present images of normal or unaltered bone histology and compare it with the effects of the use or application of YH.

3.-It turns out to be of great interest to know whether or not there are morphological changes in the bone of the specimens exposed to a hormonal decrease, compared with those that received the YH treatment.

4.-Would interesting to know what would happen in other species, for example, rabbits or sheep.

Author Response

Opinions of reviewers were attached to the article and marked in red.

Round 2

Reviewer 3 Report

The work is interesting, so I consider that it would be very valuable to have a comparative study of the changes generated by YH in bone tissue in the development of osteoporosis, presenting the morphological aspects of the possible changes (repairs) of the affected bone.